# Multigrid Beta Filter for Faster Computation of Ensemble Covariance Localization

Sho Yokota<sup>1,2</sup>, Miodrag Rancic<sup>4,3</sup>, Ting Lei<sup>4,3</sup>, R. James Purser<sup>4,3</sup>, Manuel S. F. V. De Pondeca<sup>4,3</sup>

<sup>1</sup>Numerical Prediction Development Center, Japan Meteorological Agency, Tsukuba, Ibaraki 305-0052, Japan

<sup>2</sup>Meteorological Research Institute, Japan Meteorological Agency, Tsukuba, Ibaraki 305-0052, Japan

<sup>3</sup>NOAA/NWS/NCEP/Environmental Modeling Center, College Park, Maryland 20740, USA

<sup>4</sup>Lynker, College Park, Maryland 20740, USA

Correspondence to: Sho Yokota (syokota@mri-jma.go.jp)

Abstract. This study applies a multigrid beta filter (MGBF) for covariance localization in ensemble-variational (EnVar) data assimilation instead of the conventional recursive filter (RF) to achieve faster computation in a large number of processors. The parallelization efficiency of the MGBF is higher than that of the RF because all-to-all communication to change the computational region of each processor is not necessary. However, the MGBF-based localization additionally requires horizontal variable exchange between processors; its computational cost is proportional to the number of grid points and to the ensemble size, and is generally more expensive than the RF. In this study, we implement the MGBF-based localization both for the single-scale localization and for the scale-dependent localization in the regional atmospheric EnVar data assimilation system. In addition, we clarify that applying a coarser filter grid and omitting filtering except for the coarsest resolution make the computation of the MGBF-based localization several times faster than that of the RF-based one without significantly changing the EnVar analysis.

#### 1 Introduction

In ensemble-based atmospheric data assimilation (DA), background error covariance (BEC) is one of the most important factors to determine the quality of the analysis. In general, the flow-dependent BECs created by ensemble forecasts have large sampling error for a small ensemble size. This sampling error is mitigated by the covariance localization, which decreases the ensemble-based BECs between analysis variables spatially far from each other (Hamill et al., 2001; Houtekamer and Mitchell, 2001). In ensemble-variational (EnVar, Hamill and Snyder, 2000; Lorenc, 2003) DA, however, applying the localization for all analysis variables is computationally expensive in the simplest implementation, and this cost is even more expensive when using scale-dependent localization (SDL; Buehner, 2012; Buehner and Shlyaeva, 2015) to apply large localization lengths for the long waves. Therefore, efficient calculation is an important goal to be achieved for localization.

In EnVar, the covariance localization increases the rank of the ensemble-based BEC matrix, which is attained by increasing the effective ensemble size with the Schur product of ensemble perturbations and the square root of the localization matrix

(Liu et al., 2009). Even in some other equivalent formulations of localization (e.g., Lorenc, 2003; Buehner, 2005; Bishop and Hodyss, 2009), the square root of the localization matrix is required (Ishibashi, 2015). In the simple implementation, this square root of the localization matrix is obtained by eigenvalue decomposition, where ignoring the tiny eigenvalues makes the computation faster (Liu et al., 2009).

- If the shape of localization is set to Gaussian, the square root of the localization matrix is also realized by a Gaussian filter because it is self-adjoint and its convolution is also Gaussian. Extending the earlier work of Hayden and Purser (1995) to variational analysis, Purser et al. (2003a) proposed the recursive filter (RF) as an efficient quasi-Gaussian filter applied to realize the static BEC. This RF was extended to apply to the inhomogeneous and anisotropic BEC (Purser et al., 2003b), and implemented in some operational DA systems as a method to realize the covariance localization as well as the static BEC (e.g., Wang et al., 2008, 2013; Yokota et al., 2024a). However, the RF is not necessarily parallelized efficiently when a very large number of processors are to be used because it needs to be calculated sequentially in each specific direction.
  - Purser et al. (2022) proposed another method, the multigrid beta filter (MGBF), with the potential for higher computational efficiency than the RF when using a very large number of processors for parallel computation. Unlike the RF, the MGBF is a bell-shaped filter with support of finite width, where the response is a superposition of the variables filtered at progressively coarser resolutions. Although the MGBF requires horizontal variable exchange between processors, the amount of the exchange is small in the coarser grids. Since the filter is applied for each grid, it is efficiently parallelized horizontally. It has been clarified that the MGBF makes the computation of the static BEC and the ensemble covariance localization faster (Rancic et al., 2022, 2025). However, the detail of the impact of the MGBF for the ensemble covariance localization, including SDL, has not been investigated yet.
- Based on the background above, this study applies the homogeneous isotropic MGBF for the localization, including SDL, in the regional atmospheric DA system and clarifies how to make the computation faster while keeping almost the same quality of the analysis as with the RF-based localization. Section 2 explains the formulation of the RF- and MGBF-based localizations. Section 3 describes the experimental design to clarify the impact of MGBF-based localization in the regional DA system. Section 4 discusses the results. Section 5 gives the conclusion.

## 55 2 Formulation

#### 2.1 Ensemble-variational (EnVar) data assimilation with scale-dependent localization (SDL)

This study focuses on covariance localization in the Gridpoint Statistical Interpolation (GSI)-based 3DEnVar (Wang et al., 2008, 2013). In 3DEnVar with a pure ensemble-based BEC, the analysis increment  $\delta \mathbf{x}$  is obtained by minimization of the cost function:

60 
$$J(\mathbf{a}_1, ..., \mathbf{a}_K) = \frac{1}{2} \sum_{k=1}^K (\mathbf{a}_k)^T \mathbf{L}^{-1} (\mathbf{a}_k) + \frac{1}{2} (\mathbf{H} \delta \mathbf{x} - \mathbf{d})^T \mathbf{R}^{-1} (\mathbf{H} \delta \mathbf{x} - \mathbf{d}),$$
(1)

$$\delta \mathbf{x} = \sum_{k=1}^{K} \begin{bmatrix} \mathbf{a}_k \\ \vdots \\ \mathbf{a}_k \end{bmatrix} \circ \mathbf{x}_k^{\text{en}},\tag{2}$$

where  $\mathbf{a}_k$  (k=1,...,K) is the N-dimension control vector,  $\mathbf{L}$  is the covariance localization  $(N \times N \text{ matrix})$ ,  $\mathbf{R}$  is the observation error covariance  $(M \times M \text{ matrix})$ ,  $\mathbf{H}$  is the linearized observation operator  $(M \times NV \text{ matrix})$ , and  $\mathbf{d}$  is the M-dimension observation innovation vector (K): the ensemble size; N: the number of grid points; M: the number of assimilated observations; V: the number of analysis variables).  $\mathbf{x}_k^{\text{en}}$  is the NV-dimension k-th ensemble perturbation vector (k-th ensemble member subtracted by ensemble mean and normalized by  $\sqrt{K-1}$ . In this formulation, the same localization length is applied to all analysis variables.

In applying SDL (Buehner and Shlyaeva, 2015), the analysis increment  $\delta x$  (NV-dimension vector) is obtained as:

$$\delta \mathbf{x} = \sum_{k=1}^{K} \sum_{w=1}^{W} \begin{bmatrix} \mathbf{a}_{k,w} \\ \vdots \\ \mathbf{a}_{k,w} \end{bmatrix} \circ \mathbf{x}_{k,w}^{\text{en}}, \tag{3}$$

instead of Eq. (2), where  $\mathbf{a}_k$  is extended to the *NW*-dimension vector as  $\mathbf{a}_k = \begin{bmatrix} \mathbf{a}_{k,1} \\ \vdots \\ \mathbf{a}_{k,W} \end{bmatrix}$  (*W*: the number of scales in SDL),  $\mathbf{x}_k^{\text{en}}$

is separated to multiple scales as  $\mathbf{x}_k^{\text{en}} = \sum_{w=1}^W \mathbf{x}_{k,w}^{\text{en}}$ , and  $\mathbf{L}$  is extended to the  $NW \times NW$  matrix as:

$$\mathbf{L} = \begin{bmatrix} \mathbf{L}_{1}^{1/2} & \mathbf{0} \\ & \ddots & \\ \mathbf{0} & \mathbf{L}_{W}^{1/2} \end{bmatrix} \mathbf{E} \begin{bmatrix} \mathbf{L}_{1}^{T/2} & \mathbf{0} \\ & \ddots & \\ \mathbf{0} & \mathbf{L}_{W}^{T/2} \end{bmatrix}, \tag{4}$$

where  $\mathbf{L}_w$  is the  $N \times N$  localization matrix applied for w-th scale of ensemble perturbations  $\mathbf{x}_{k,w}^{\mathrm{en}}$ , and  $\mathbf{E} = \begin{bmatrix} \mathbf{I} & \cdots & \mathbf{I} \\ \vdots & \ddots & \vdots \\ \mathbf{I} & \cdots & \mathbf{I} \end{bmatrix}$  is the

NW × NW matrix to combine each scale for localizing cross-scale covariances in SDL ("Cross" in Huang et al., 2021).

#### 75 2.2 Recursive filter (RF)-based localization

65

The calculation of the localization  $\mathbf{L}$  is accomplished by the RF (Purser et al., 2003a) in the GSI-based 3DEnVar as shown in Fig. 1a, where the square root of the localization matrix  $\mathbf{L}_w \left( = \mathbf{L}_w^{1/2} \mathbf{L}_w^{T/2} \right)$  is quasi-Gaussian and computed as:

$$\mathbf{L}_{w}^{1/2} = \mathbf{F}_{\mathrm{RF}}^{Z} \mathbf{F}_{\mathrm{RF}}^{Y} \mathbf{F}_{\mathrm{RF}}^{X}. \tag{5}$$

 $\mathbf{F}_{RF}^{X}$ ,  $\mathbf{F}_{RF}^{Y}$ , and  $\mathbf{F}_{RF}^{Z}$  denote RFs in x-, y-, and z-directions, respectively (self-adjoint  $N \times N$  matrices). These RFs should be

applied recursively; for example, to obtain  $\begin{bmatrix} \mathbf{q}_{1}^{\text{out}} \\ \vdots \\ \mathbf{q}_{N_{X}}^{\text{out}} \end{bmatrix} = \mathbf{F}_{\text{RF}}^{X} \begin{bmatrix} \mathbf{q}_{1}^{\text{in}} \\ \vdots \\ \mathbf{q}_{N_{X}}^{\text{in}} \end{bmatrix},$ 

$$\mathbf{q}_{i}^{\text{mid}} = \beta \mathbf{q}_{i}^{\text{in}} + \sum_{i=1}^{p} \alpha_{i} \mathbf{q}_{i-i}^{\text{mid}}$$

$$\tag{6}$$

is sequentially calculated from the smallest i, and after that,

$$\mathbf{q}_{i}^{\text{out}} = \beta \mathbf{q}_{i}^{\text{mid}} + \sum_{j=1}^{p} \alpha_{j} \mathbf{q}_{i+j}^{\text{out}}$$
(7)

is sequentially calculated from the largest i, where  $\mathbf{q}_i^{\text{in}}$ ,  $\mathbf{q}_i^{\text{mid}}$  and  $\mathbf{q}_i^{\text{out}}$  ( $i = 1, ..., N_X$ ) are  $N_Y N_Z$ -dimension vectors ( $N_X$ ,  $N_Y$ , and  $N_Z$  are the numbers of grid points in x-, y-, and z-directions, respectively, so  $N = N_X N_Y N_Z$ ).  $\mathbf{q}_{i-j}^{\text{mid}}$  ( $i - j \le 0$ ) and  $\mathbf{q}_{i+j}^{\text{out}}$  ( $i + j \ge N_X$ ) are zero. The coefficients  $\beta$  and  $\alpha_j$  (j = 1, ..., p; p is the order of RF) are set to make the filtering kernel of  $\mathbf{F}_{\text{RF}}^X$  quasi-Gaussian as:

$$G_p(x) \xrightarrow[p \to \infty]{} c_G \exp\left(-\frac{x^2}{s^2}\right),$$
 (8)

where the coefficient  $c_G$  is set to satisfy  $\int_{-\infty}^{\infty} [G_p(x)]^2 dx = 1$ . Since the resulting filtering kernel of  $\mathbf{F}_{RF}^X (\mathbf{F}_{RF}^X)^T$  is the self-convolution of  $G_p(x)$  as:

$$G_p * G_p(x) \equiv \int_{-\infty}^{\infty} G_p(x - x') G_p(x') dx' \underset{p \to \infty}{\longrightarrow} \exp\left(-\frac{x^2}{2s^2}\right), \tag{9}$$

s is the standard deviation of  $G_{\infty} * G_{\infty}(x)$ , which is the same as the  $e^{-1/2}$ -folding scale  $\sigma$ .

Since Eqs. (6) and (7) are calculated sequentially, RF in one-direction is efficiently parallelized only in the other direction; for example,  $\mathbf{F}_{RF}^X$  is efficiently parallelized only for  $N_Y N_Z$  and the parallelization for  $N_X$  is impossible. Therefore, all-to-all communication to change the direction of parallelization, which degrades the parallelization efficiency with the large number of processors, is required to calculate  $\mathbf{F}_{RF}^Z \mathbf{F}_{RF}^Y \mathbf{F}_{RF}^X$  (e.g., between  $\mathbf{F}_{RF}^Z$  and  $\mathbf{F}_{RF}^Y \mathbf{F}_{RF}^X$ ). Note that  $\mathbf{L}$  itself is also calculated in parallel for the ensemble size K considering the formulation in Eq. (1).

#### 2.3 Multigrid beta filter (MGBF)-localization

This study suggests to calculate the localization  $\mathbf{L}$  with MGBF instead of RF. Although the original MGBF (Purser et al., 2022) superposes variables filtered in filter grids of multiple resolutions  $\mathbf{g}_t$  (t = 1, ..., T; the grid interval of  $\mathbf{g}_{t+1}$  is twice coarser than  $\mathbf{g}_t$ ), this study applies MGBF only for the coarsest filter grid  $\mathbf{g}_T$  for faster computation as shown in Fig. 1b, where  $\mathbf{L}_w^{1/2}$  is computed as:

$$\mathbf{L}_{w}^{1/2} = \mathbf{D}_{\mathbf{g}_{0} \leftarrow \mathbf{g}_{1}} \mathbf{F}_{\mathrm{BF}(\mathbf{g}_{1})}^{Z} \mathbf{D}_{\mathbf{g}_{1} \leftarrow \mathbf{g}_{T}} \mathbf{F}_{\mathrm{BF}(\mathbf{g}_{T})}^{Y} \mathbf{F}_{\mathrm{BF}(\mathbf{g}_{T})}^{X} \mathbf{D}_{\mathbf{g}_{T} \leftarrow \mathbf{g}_{1}}, \tag{10}$$

where  $\mathbf{D}_{g_T \leftarrow g_1}$  ( $N_{g_T} \times N_{g_1}$  matrix) is 2x2-points bilinear interpolations with doubling the coefficients to satisfy  $\mathbf{D}_{g_T \leftarrow g_1} \mathbf{D}_{g_T \leftarrow g_1}^T = \mathbf{I}$ , which is repeated from  $\mathbf{g}_1$  (the finest filter grid) to  $\mathbf{g}_T$  (the coarsest filter grid),  $\mathbf{D}_{g_1 \leftarrow g_T}$  ( $N_{g_1} \times N_{g_T}$  matrix) is linearly weighted biquadratic horizontal interpolations (down-sending) repeated from  $\mathbf{g}_T$  to  $\mathbf{g}_1$ , and  $\mathbf{D}_{g_0 \leftarrow g_1}$  ( $N \times N_{g_1}$  matrix) is bilinear horizontal and vertical interpolations (mapping) from  $\mathbf{g}_1$  to the analysis grid  $\mathbf{g}_0$  ( $N_{g_t}$ : the number of grid points in  $\mathbf{g}_t$ ). The finest filter grid  $\mathbf{g}_1$  is the same as the analysis grid  $\mathbf{g}_0$  or coarser. Note that  $\mathbf{D}_{g_T \leftarrow g_1}$  is required only in SDL because  $\mathbf{D}_{g_T \leftarrow g_1} \mathbf{E} \mathbf{D}_{g_T \leftarrow g_1}^T = \mathbf{I}$  in single-scale localization.  $\mathbf{F}_{BF(g_t)}^X$ ,  $\mathbf{F}_{BF(g_t)}^Y$ , and  $\mathbf{F}_{BF(g_t)}^Z$  denote isotropic line beta filters applied in each generation in x-, y-, and z-directions, respectively (self-adjoint  $N_{g_t} \times N_{g_t}$  matrices); for example, the filtering kernel of  $\mathbf{F}_{BF(g_t)}^X$  is:

$$B_{p,t}(x) = c_{B,t}(1 - 4X^2)^p \quad \left( X \equiv \frac{|x|}{s\sqrt{2(2p+3)}} \le \frac{1}{2} \right)$$
 (11)

where  $B_{p,t}(x) = 0$  in X > 1/2, the coefficient  $c_{B,t}$  is set to satisfy  $\int_{-\infty}^{\infty} [B_{p,t}(x)]^2 dx = \omega_t$  ( $\omega_t$ : weight of  $g_t$  where  $\sum_{t=1}^T \omega_t = \omega_T = 1$ ), and s is the standard deviation of the self-convolution of  $B_{p,t}(x)$ , which is the filtering kernel of

$\mathbf{F}_{\mathrm{BF}(\mathbf{g}_t)}^{X}(\mathbf{F}_{\mathrm{BF}(\mathbf{g}_t)}^{X})^{T}$  and can be shown to have the form:

$$B_{p,t} * B_{p,t}(x) = \omega_t (1 - X)^{2p+1} \sum_{i=0}^p a_{i,p} X^{p-i} (1 + X)^{2i} \quad (X \le 1).$$
 (12)

If we generalize the definition of binomial coefficients:

$$C(i,j) = \frac{i!}{(i-j)!j!},\tag{13}$$

then the coefficients can be expressed,

$$a_{i,p} = \sum_{j=0}^{\min(i,\lfloor (p-i)/2\rfloor)} \frac{C(p,j)C(p-i,2j)C(i,j)}{C(2p,2j)},$$
 (14)

where  $[\cdot]$  is the floor function. In the particular case, p=2, these coefficients are  $a_{0,2}=a_{1,2}=a_{2,2}=1$ . The filtering kernel of  $\mathbf{F}_{\mathrm{BF}(\mathbf{g}_T)}^X (\mathbf{F}_{\mathrm{BF}(\mathbf{g}_T)}^X)^T$  obtained as the self-convolution of  $B_{2,T}(x)$  can be expanded as:

$$B_{2,T} * B_{2,T}(x) = (1 - X)^5 (1 + 5X + 9X^2 + 5X^3 + X^4) \quad \left( X \equiv \frac{|x|}{s\sqrt{14}} \le 1 \right), \tag{15}$$

where s is the standard deviation of  $B_{2,T} * B_{2,T}(x)$ . Unlike RF, s is smaller than the  $e^{-1/2}$ -folding scale  $\sigma$  in MGBF (here,  $s/\sigma \sim 0.92852$ ).

In MGBF, not only  $\mathbf{F}_{\mathrm{BF}(\mathbf{g}_1)}^Z$  but also  $\mathbf{F}_{\mathrm{BF}(\mathbf{g}_T)}^X$  and  $\mathbf{F}_{\mathrm{BF}(\mathbf{g}_T)}^Y$  are parallelized for  $N_X N_Y$  because Eq. (11) is independently applied for each horizontal grid point only in the finite domain near the point. It indicates that communication between processors is limited to the exchange of halo grid points with spatially neighboring processors and all-to-all communication is not required in MGBF.

(b) (a) cost function nterpolation interpolation  $\mathbf{D}_{\sigma_0 \leftarrow \sigma_s}$ (mapping) Minimizing Vertical beta filter  $\mathbf{F}_{\mathrm{BF}(\mathrm{g}_{1})}^{z}$ Vertical beta filter  $\mathbf{F}_{\mathrm{BF}(\mathrm{g}_{1})}^{Z}$ cost function Vertical Vertical rsive filter recursive filter Forward Adjoint Recursive Multigrid All-to-all All-to-all interpolation interpolation (up-sending)  $\mathbf{D}_{g_1 \leftarrow g_T}^T$ communication communication down-sending) filter beta filter Horizontal Horizontal recursive filter recursive filter  $F_{RF}^{Y}F_{RF}^{X}$  $F_{RF}^X F_{RF}^Y$ Horizontal beta filter beta filter  $\mathsf{F}_{\mathrm{BF}(\mathsf{g}_T)}^{Y} \mathsf{F}_{\mathrm{BF}(\mathsf{g}_T)}^{X}$  $F_{BF(g_T)}^X F_{BF(g_T)}^Y$ Forward 4-points linear 4-points linear terpolation interpolation  $\mathbf{D}_{g_T \leftarrow g_1}^T$ 

Figure 1: Schematics of procedures of (a) RF and (b) MGBF.

## 3 Experimental design

To compare the computation time and the 3DEnVar analysis between RF- and MGBF-based localizations, this study conducted hourly analysis-forecast cycling experiments. The experiments consist of GSI-based pure 3DEnVar and the limited area model capability for the non-hydrostatic finite-volume cubed-sphere dynamical core (FV3LAM, Lin, 2004; Putman and Lin, 2007; Black et al., 2021) in a prototype Rapid Refresh Forecast System (RRFS, Carley et al., 2023) in National Centers for Environmental Prediction (NCEP). The FV3LAM applied physics schemes listed in Table 1, and covered the CONUS (contiguous United States) domain with the horizontal grid interval of 3 km, where the number of grid points in *x*-, *y*-, and *z*-directions are (1820,1092,65). The lowest level thickness and the top of the model are 8 m and 2 hPa, respectively. In 3DEnVar, the number of analysis grid points were set to (*N<sub>X</sub>*,*N<sub>Y</sub>*,*N<sub>Z</sub>*) = (910,546,65); namely the horizontal grid interval was twice as large as that of the FV3LAM. The larger interval of the analysis grid reduces the computational cost but makes the resolution of analysis increments coarser and prevents to set the localization length smaller than the grid interval.

Figure 2 shows the schematics of the sensitivity experiments. The selected experimental period includes when Hurricane Ian moved from the area northeast of Florida toward South Carolina (Bucci et al., 2023). All cycling experiments started from the same 1 h FV3LAM deterministic forecast initiated with the pure 3DEnVar analysis at 15 UTC, 29 September 2022, where the first guess as the initial condition (IC) was the 3 h forecast in the Global Forecast System (GFS, horizontal grid interval ~ 13 km) in NCEP, and ensemble BEC was created by the 9 h 80 member global ensemble forecasts in the Global DA System (GDAS, horizontal grid interval ~ 26 km) in NCEP. After that, hourly analysis-forecast cycles with pure 3DEnVar and FV3LAM forecasts were repeated until 00 UTC, 30 September.

All ensemble BECs for the pure 3DEnVar analyses except at 15 UTC were created by ensemble analysis-forecast cycles [30 member hourly FV3LAM ensemble forecasts and serial ensemble square root filter (EnSRF; Whitaker and Hamill, 2002)] initiated with the 9 h ensemble forecast subset (first 30 of 80 members) at 15 UTC in the GDAS. The cutoff lengths of the Gaspari-Cohn localization function (Gaspari and Cohn, 1999) in EnSRF were set to 300 km horizontally and 1.1 scale heights vertically. After each EnSRF analysis (just before the next ensemble forecasts), the ensemble mean was replaced with the variational analysis (recentering in Fig. 2) and the ensemble spread was inflated by the relaxation-to-prior spread method (RTPS; Whitaker and Hamill, 2012) with a factor of 0.85.

Both deterministic and ensemble analysis-forecast cycles adopted the GFS forecasts as the lateral boundary conditions (LBCs), and assimilated observations associated with the Rapid Refresh (RAP; Benjamin et al., 2004, 2016) from METAR, rawinsondes, aircraft, and radial winds of Weather Surveillance Radar-1988 Doppler (WSR-88D; Crum and Alberty, 1993, Liu et al., 2016). Although satellite radiance, radar reflectivity, and lightning data were not assimilated directly, they were used in land-snow DA (Benjamin et al., 2022) and non-variational cloud analysis (Benjamin et al., 2021) to correct hydrometeors, temperature, and specific humidity after each 3DEnVar analysis (just before the next deterministic forecasts).

The only difference among sensitivity experiments is how to apply the localization for pure 3DEnVar (Table 2). In RF, the RF-based single-scale localization (W=1; p=2; localization length s: 82.158 km horizontally and 3 grids vertically) was applied. In MGBF00–04, the RF-based horizontal localization in RF was replaced to the MGBF-based one with the same localization length s and the exponent p as that in RF. In MGBF00–02, the number of finest filter grids  $N_{\rm g_1}$  was the same as that of analysis grid, where BF was applied for the finest grid  ${\rm g_1}$  in MGBF00 but the coarser grid  ${\rm g_4}$  in MGBF01–02. In MGBF03–04, filter grids for  ${\rm g_1}$  were horizontally coarser ( $N_{\rm g_1}$  was smaller) than those in MGBF00–02 and the filter was applied for  ${\rm g_2}$ . In MGBF04, the filter grids were coarser also vertically, and RF-based vertical localization was replaced to MGBF-based one in addition to the horizontal localization. The MGBF04 $\sigma$  is the same as MGBF04 except with the smaller localization length s, which was decreased by the factor of 0.92852 to make the  $e^{-1/2}$ -folding scale  $\sigma$  the same as that in RF. RFSDL, MGBF03SDL, MGBF04SDL, and MGBF04 $\sigma$ SDL are the same as RF, MGBF03, MGBF04, and MGBF04 $\sigma$ , respectively, except for applying fourfold horizontal localization lengths additionally as larger-scale SDL (W=2). In all MGBF-based localizations,  ${\rm g_t}$  (t=2,..., T) was calculated in parallel after  ${\rm g_1}$ . Since the calculation of  ${\rm g_1}$  is meaningless in case the weight for  ${\rm g_1}$  set to zero, it was skipped for faster computation except in MGBF00–01. The number of processors for the parallel computation was set to 735 (35 in the x-direction and 21 in the y-direction) for all experiments. Note that only the first pure 3DVar analysis at 15 UTC applied the same localization as in RF for all experiments.

Table 2: List of physics schemes used in FV3LAM.

| Physics schemes                   | Specification  Thompson-Eidhammer Aerosol Aware Microphysics (Thompson and Eidhammer 2014)                                             |  |  |  |  |
|-----------------------------------|----------------------------------------------------------------------------------------------------------------------------------------|--|--|--|--|
| Cloud microphysics                |                                                                                                                                        |  |  |  |  |
| Planetary boundary layer          | Mellor-Yamada-Nakanishi-Niino Eddy Diffusivity/Mass Flux (MYNN-EDMF; Nakanish and Niino 2009; Olson et al. 2019; Angevine et al. 2020) |  |  |  |  |
| Surface layer                     | Mellor-Yamada-Nakanishi-Niino (MYNN) surface layer (Olson et al. 2021)                                                                 |  |  |  |  |
| Gravity wave                      | Small Scale Gravity Wave Drag (SSGWD; Tsiringakis et al. 2017) and Turbulent Orographic Form Drag (TOFD; Beljaars et al. 2004)         |  |  |  |  |
| Land                              | Rapid Update Cycle Land Surface Model (RUC LSM; Smirnova et al. 1997, 2000, 2016)                                                      |  |  |  |  |
| Long and short-<br>wave radiation | Rapid Radiative Transfer Model for Global Circulation Models (RRTMG; Mlawer et al. 1997; Iacono et al. 2008)                           |  |  |  |  |

 $GFS9h\ GFS10h\ GFS11h\ GFS12h\ GFS7h\ GFS8h\ GFS9h\ GFS10h\ GFS11h\ GFS12h$  forecast foreca

Figure 2: Schematics of analysis-forecast cycles with the RRFS.

Table 2: List of localization settings for pure 3DEnVar in sensitivity experiments.

| Name       | Horizontal<br>filter                       | Vertical<br>filter  | Number of the finest filter grids $N_{g_1}$ | Weight $(\omega_1, \omega_2, \omega_3, \omega_4)$ ("-" indicates no filtering) | Horizontal localization length <i>s</i> (km) | Vertical localization length <i>s</i> (grid unit) |
|------------|--------------------------------------------|---------------------|---------------------------------------------|--------------------------------------------------------------------------------|----------------------------------------------|---------------------------------------------------|
| RF         | RF                                         | RF                  | -                                           | -                                                                              | 82.158                                       | 3.0000                                            |
| MGBF00     | $BF(g_1)$                                  | RF                  | (910,546,65)                                | (1,0,-,-)                                                                      | 82.158                                       | 3.0000                                            |
| MGBF01     | $BF(g_4)$                                  | RF                  | (910,546,65)                                | (0,0,0,1)                                                                      | 82.158                                       | 3.0000                                            |
| MGBF02     | $BF(g_4)$                                  | RF                  | (910,546,65)                                | (-,0,0,1)                                                                      | 82.158                                       | 3.0000                                            |
| MGBF03     | $BF(g_2)$                                  | RF                  | (280,168,65)                                | (-,1,-,-)                                                                      | 82.158                                       | 3.0000                                            |
| MGBF04     | $BF(g_2)$                                  | $BF(g_1)$           | (280,168,33)                                | (-,1,-,-)                                                                      | 82.158                                       | 3.0000                                            |
| MGBF04σ    | $BF(g_2)$                                  | $BF(g_1)$           | (280,168,33)                                | (-,1,-,-)                                                                      | 76.286                                       | 2.7856                                            |
| RFSDL      | RF<br>RF                                   | RF<br>RF            | -                                           | -                                                                              | 328.63<br>82.158                             | 3.0000<br>3.0000                                  |
| MGBF03SDL  | $ BF(g_4) $ $ BF(g_2) $                    | RF<br>RF            | (280,168,65)<br>(280,168,65)                | (-,0,0,1)<br>(-,1,-,-)                                                         | 328.63<br>82.158                             | 3.0000<br>3.0000                                  |
| MGBF04SDL  | BF(g <sub>4</sub> )<br>BF(g <sub>2</sub> ) | $BF(g_1)$ $BF(g_1)$ | (280,168,33)<br>(280,168,33)                | (-,0,0,1)<br>(-,1,-,-)                                                         | 328.63<br>82.158                             | 3.0000<br>3.0000                                  |
| MGBF04σSDL | $ BF(g_4) $ $ BF(g_2) $                    | $BF(g_1)$ $BF(g_1)$ | (280,168,33)<br>(280,168,33)                | (-,0,0,1)<br>(-,1,-,-)                                                         | 305.14<br>76.286                             | 2.7856<br>2.7856                                  |

## 4 Results and discussion

## 4.1 Single observation data assimilation

In this subsection, the filter responses of the RF- and MGBF-based localizations are compared with single pseudo-observation DA. Here, a single surface pressure observation was assimilated with –10 hPa innovation and 1 hPa observation error in the northern region of Hurricane Ian at 80° W and 31° N, where the first guess was the 1 h FV3LAM forecast at 16 UTC, 29 September.

Figure 3 shows analysis increments of sea-level pressure (SLP). Compared to the increments with the single-scale localization (Figs. 3a,b), the SDL created the larger scale flow-dependent increments both for the RF- and MGBF-based localizations (Figs. 3c,d) since the horizontal localization length in the larger-scale SDL was set to fourfold. The difference between the RF- and MGBF-based localizations was little compared to the difference between the single-scale localization and SDL.

To clarify the difference of the responses between the RF- and MGBF-based localizations in more detail, the meridional cross-section of the ratio of analysis increments with and without the localization (analysis increments in RF, MGBF00, MGBF04, and MGBF04σ divided by the increment without the localization), which are regarded as the filter responses of each experiment, are shown in Fig. 4. While the response of RF (cyan line) was almost the same as Gaussian, that of MGBF00 (brown line) was a little wider, and almost consistent with Eq. (15). The difference between MGBF00 (brown line) and MGBF04 (pink line) was hardly visible although it was slightly underestimated near the peak in MGBF04 due to the coarser filter grid. Compared to MGBF04 (pink line), the response of MGBF04σ (yellow line) was closer to Gaussian near the *e*<sup>-1/2</sup>-folding scale while it was smaller far from the observation.

Figure 3: Analysis increment (color, hPa) and analysis (gray contours, every 4 hPa) of SLP at 16 UTC, 29 September 2022 in the single surface pressure DA experiments (a: RF; b: MGBF04; c: RFSDL; d: MGBF04SDL). Yellow dot is the position of the assimilated observation.

Figure 4: Meridional cross-section of analysis increment of SLP at 16 UTC, 29 September 2022 in the single surface pressure DA experiments (cyan: RF; brown: MGBF00; pink: MGBF04; yellow: MGBF04 $\sigma$ ) divided by that without spatial localization. The black dashed line is Gaussian and the other dashed lines are the differences from Gaussian. The horizontal dotted line is  $e^{-0.5}$  (~0.60653) and the vertical dotted line is the  $e^{-0.5}$ -folding length of Gaussian (~82.158 km), respectively.

## 220 4.2 Analysis-forecast cycling experiments

In this subsection, the calculation time of the RF- and MGBF-based localizations and the qualities of the resulting analyses are compared. Figure 5 shows the computation times for localizations in analysis-forecast cycling experiments. The time for horizontal filtering in MGBF01–02 was smaller than that of MGBF00 because it was applied in the coarser filter grid  $g_4$ ; in MGBF02, it was about half of that in MGBF01 due to skipping the filter for  $g_1$ . However, the total time for the localization in MGBF00–02 was larger than that in RF because the amount of the calculation and communication between processors in up-sending and down-sending were proportional to the number of grid points, which were large in MGBF00–02. On the other hand, the time for the localization in MGBF03–04, which applied a coarser  $g_1$  than MGBF00–02, was shorter than that in RF. In particular, the time for the localization in MGBF04, which applied vertical MGBF in the coarser vertical grid, was

about 20 % of that in RF. In SDL, the total time for the localization was roughly twice that of single-scale localization both for the RF- and MGBF-based localizations, which means that the reduction of the computation time by the MGBF-based localization was also approximately twice in SDL. The reduction rate of the computation time by the MGBF-based localization was larger in the experiments with larger numbers of processors (not shown), which indicates that parallelization efficiency of the MGBF is higher than that of the RF including the all-to-all communication. Hereafter, only RF, MGBF04, MGBF04σ, RFSDL, MGBF04SDL, and MGBF04σSDL are focused to show the small difference of the analyses with computationally efficient MGBF from that with RF.

Despite the large reduction of the computation time, the difference of analysis increments of SLP between the RF- and MGBF-based localizations was small in both experiments with the single-scale localization and the SDL (Fig. 6). The relatively large difference near Hurricane Ian (Figs. 6c–f) is reasonable due to the large increment there (Figs. 6a, b). In the experiments with SDL, the difference is slightly larger in the maritime area (Figs. 6d, f) probably because the difference between RF and MGBF is more obvious in the large localization applied to the large-scale ensemble-based error covariance, which is also large in the maritime area. Note that the analysis increment was not spatially smoothed even in the MGBF-based localization with the coarse filter grid because the ensemble perturbations  $\mathbf{x}_{k,w}^{\text{en}}$  in Eq. (3) was not affected by the MGBF. Moreover, the difference from RF (Fig. 6a) was slightly smaller in MGBF04 $\sigma$ CFig. 6e) than that in MGBF04 (Fig. 6b) was also slightly smaller in MGBF04 $\sigma$ SDL (Fig. 6f) than that in MGBF04SDL (Fig. 6d) even though the computation times for MGBF04 $\sigma$ CDL were almost the same as that for MGBF04 and MGBF04SDL, respectively (not shown).

The impact of the MGBF-based localization on the dynamical balance of the analysis was also small. Figure 7 shows the mean absolute pressure tendency of the forecast from the analysis at 16 UTC, 29 September. While it was smaller in the experiments with SDL than that with single-scale localization (consistent with Yokota et al., 2024b), the impact of the MGBF-based localization was relatively small; for example, the difference between RF (cyan line) and MGBF04 (pink line) was smaller than that between RF (cyan line) and RFSDL (blue line). However, this slight difference between the RF- and MGBF-based localizations was accumulated in the analysis-forecast cycle, and the pressure tendency of the forecast from the last analysis with the MGBF-based localization at 00 UTC, 30 September was larger than that with the RF-based localization (Fig. 8) probably because the MGBF was the compact-support filter and its filter response was limited to the finite region. Nevertheless, MGBF04σ (yellow line) showed a smaller deviation from RF (cyan line) than MGBF04 (pink line). Similarly, MGBF04σSDL (orange line) was closer to RFSDL (blue line) than MGBF04SDL (red line).

Figure 9 shows the first guess departure of assimilated in-situ temperature and horizontal wind observations in the whole analysis-forecast cycles. For temperature, the RMSE and cold bias in the experiments with SDL were smaller than those with single-scale localization (consistent with Yokota et al., 2024b), and the differences between the RF- and MGBF-based localizations were relatively small (Figs. 9a and b). For horizontal wind, on the other hand, the degradation of the RMSE by the MGBF-based localization (pink line in Fig. 9c) were not necessarily smaller than the improvement by the SDL (blue line in Fig. 9c) probably because the impact of SDL on horizontal wind was smaller than that on temperature. However, the

difference from RF (cyan line) was smaller in MGBF04σ (yellow line) than that in MGBF04 (pink line), and the difference from RFSDL (blue line) was also smaller in MGBF04σSDL (orange line) than that in MGBF04SDL (red line).

Considering the results above, the quality of the analysis in RF and RFSDL was closer to that in MGBF04 $\sigma$  and MGBF04 $\sigma$ SDL than that in MGBF04 and MGBF04SDL, respectively. It may indicate that the  $e^{-0.5}$ -folding scale of the localization function  $\sigma$  is more sensitive to the quality of the analysis than the standard deviation s. Note that these differences of the analyses discussed here hardly affected the Hurricane Ian forecasts. In fact, the track forecasts and associated precipitation forecasts initiated with the last analyses with the MGBF-based localization at 00 UTC, 30 September were almost the same as those with the RF-based localization (Figs. 10 and 11a). The minimum SLP forecasts with the RF-and MGBF-based localizations were also almost the same and the differences were smaller than that with and without SDL (Fig. 11b).

Figure 5: Computation time for localization [green: vertical filtering (mapping between analysis and filter grids is included for MGBF); blue: all-to-all communication (only for RF); orange: up-sending and down-sending between generations (only for MGBF); red: horizontal filtering (weighting is included for MGBF)] averaged from 16UTC, 29 September to 00UTC, 30 September 2022 in each experiment. Error bars show minimum and maximum.

Figure 6: Analysis increment (color, hPa) and first guess (gray contours, every 4 hPa) of SLP at 16UTC, 29 September 2022, in (a) RF and (b) RFSDL, and difference of the SLP analysis (hPa) from RF or RFSDL (c: MGBF04–RF; d: MGBF04SDL–RFSDL; e: MGBF04σ–RF; f: MGBF04σSDL–RFSDL).

Figure 7: Mean absolute pressure tendency (hPa  $h^{-1}$ ) of the 1 h forecasts from the analysis at 16 UTC, 29 September 2022 in each experiment (cyan: RF; pink: MGBF04; yellow: MGBF04 $\sigma$ ; blue: RFSDL; red: MGBF04SDL; orange: MGBF04 $\sigma$ SDL).

Figure 8: Same as Fig. 7 except for the first 1 h forecasts from the analysis at 00 UTC, 30 September 2022.

Figure 9: Vertical profiles of first guess departure (a,c) standard deviations (difference from RF) and (b,d) biases verified against assimilated in-situ observations [a,b: temperature (K); c,d: horizontal wind (m s<sup>-1</sup>)] in each cycling experiment (cyan: RF; pink: MGBF04; yellow: MGBF04σ; blue: RFSDL; red: MGBF04SDL; orange: MGBF04σSDL) from 15 UTC, 29 September to 00 UTC, 30 September 2022. Square marks indicate significantly different from RF (confidence level ≥ 95 % in the t-test). The cyan lines are not shown in (a) and (c) and are almost superposed by the pink and yellow lines in (b) and (d).

Figure 10: Composited radar reflectivity (color, dBZ) and SLP (blue contours, every 4 hPa) analyses at 00UTC, 30 September 2022, and Hurricane Ian track forecasts (black lines) in each experiment (a: RF; b: MGBF04; c: MGBF04σ; d: RFSDL; e: MGBF04SDL; f: MGBF04σSDL) and (g) Multi-Radar Multi-Sensor (MRMS; Smith et al., 2016) composite reflectivity and High-Resolution Rapid Refresh (HRRR; Dowell et al., 2022) SLP analysis. White lines are Ian's best track.

Figure 11: (a) Location error verified against the best track (km) and (b) minimum SLP (hPa) of Hurricane Ian forecasts initialized at 00UTC, 30 September 2022, in each experiment (cyan: RF; pink: MGBF04; yellow: MGBF04σ; blue: RFSDL; red: MGBF04SDL; orange: MGBF04σSDL). Black dotted line in (b) indicates the best track.

#### 5 Conclusions

This study applied the MGBF for the ensemble covariance localization instead of the RF in the regional EnVar DA system, and showed how to make the computation faster than the RF. If the analysis grid was mapped to the coarser filter grid and the filter was applied only in the grid with the coarsest resolution, the MGBF sped the computation of the localization (approximately by five times with 735 processors) without a significant degradation of the quality of the analysis, both for the single-scale localization and for the SDL (Fig. 5). Note that the analysis increment was not spatially smoothed even in the MGBF-based localization with the coarse filter grid.

Since this study applied the MGBF only on the grid with the coarsest resolution, the filter response was the convolution of the strict beta function [Eq. (15) and Fig. 4]. Unlike RF, the  $e^{-0.5}$ -folding scale of this function was larger than the standard deviation, which caused the small difference of the quality of the analysis between the RF- and MGBF-based localizations.

- However, this difference was mitigated by applying the smaller localization length for the MGBF to make the  $e^{-1/2}$ -folding scale the same as that in RF (Figs. 6–9). An alternative would be to replace the simple beta filter with the 'tri-beta' line filter recently proposed by Purser (2024), which produces a profile more closely conforming to the intended Gaussian.
  - The idea to apply the compact-support filter with the coarse resolution is the same as the Normalized Interpolated Convolution from an Adaptive Subgrid (NICAS) adopted in the Model for Prediction Across Scales-Atmosphere with the Joint Effort for Data assimilation Integration (JEDI-MPAS, Liu et al. 2022). The NICAS applies a localization matrix on the unstructured coarse filter grid and interpolates it to the analysis grid directly. On the other hand, the MGBF-based localization applies a filter on the structured coarse filter grid and interpolates it from the coarsest filter grid  $g_T$  to the analysis grid  $g_0$  step by step. One advantage of the MGBF-based localization is high parallelization efficiency with the step-by-step interpolation. However, note that the computational cost of the analysis with small localization length in MGBF is not necessarily smaller than that in RF since the interval of the filter grid should be smaller than the localization length.
  - Despite the small difference of the analysis between the RF- and MGBF-based localizations, it may be significant after many analysis-forecast cycles since the impact of the compact-support MGBF is accumulated (Fig. 8). To make the MGBF-based localization further closer to the RF-based one, it may be required to apply the MGBF also in the grid with the finer resolution and calibrate the localization length and the weight of each resolution.
- This study showed similarity of RF and MGBF only in the single case. However, the small difference even in the case of the strong Hurricane implies the much smaller difference in general cases. The longer cycling test for more reliable verification is the future task since it requires huge computational resources.
  - This study focused only on the computational efficiency of the homogeneous isotropic MGBF. However, the advantages of the MGBF compared to the RF are not only the computational efficiency but also the flexible settings for various filter responses including inhomogeneity and anisotropy (Purser et al., 2022). To make the shape of localization more sophisticated within the MGBF is also one of the important future tasks to be carried out.

## Code and data availability

ICs, LBCs, and unrestricted observation data used in this study are obtained from https://doi.org/10.5281/zenodo.15744386 (NOAA National Centers for Environmental Prediction, 2025a), https://doi.org/10.5281/zenodo.15747450 (NOAA National 345 Centers for Environmental Prediction, 2025b), and https://doi.org/10.5281/zenodo.15747477 (NOAA National Centers for Prediction, 2025c). **RRFS** obtained Environmental The system used in this study is from https://doi.org/10.5281/zenodo.15193112 (Yokota, 2025).

## **Author contributions**

SY designed data assimilation experiments and the verifications, carried them out, and wrote the original manuscript; SY, MR, TL, and JP developed the data assimilation system; MR, TL, JP, and MP reviewed the manuscript.

#### **Competing interests**

The authors declare that they have no conflict of interest.

## Acknowledgements

The authors thank RRFS developers in the Environmental Modeling Center and the Global Systems Laboratory for setting
up the experiments with the RRFS, and Samuel Degelia and Gang Zhao for their thoughtful reviews on an earlier version of
this manuscript. This study is supported by the National Weather Service Office of Science and Technology Integration
through the University Corporation for Atmospheric Research (UCAR) Cooperative Programs for the Advancement of Earth
System Science (CPAESS), the NOAA Research and Development High Performance Computing Program, and Mississippi
State University's High Performance Computing System. We used one of the NOAA Research and Development High
Performance Computing Systems (RDHPCS), ORION, located at Mississippi State University for conducting the
experiments in this study. This research of the National Centers for Environmental Prediction (NCEP) Environmental
Modeling Center (EMC) is supported by NOAA's Science Collaboration Program and administered by UCAR's Cooperative
Programs for the Advancement of Earth System Science (CPAESS) under award #NA21OAR4310383 and
NA23OAR4310383B.

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
