# Peer review of "Multigrid Beta Filter for Faster Computation of Ensemble Covariance Localization"

_EGUsphere, 2025_

## Author Response (AR1)

**RESPONSE TO EDITOR:**

We changed the following parts in the manuscript in addition to the revision based on referees' comments.

**Response 1:**

We additionally cited Rancic et al. (2025), which was recently published and closely related to our manuscript.

**Changes in manuscript:**

It has been clarified that the MGBF makes the computation of the static BEC and the ensemble covariance localization faster (Rancic et al., 2022, 2025). However, the detail of the impact of the MGBF for the ensemble covariance localization, including SDL, has not been investigated yet. (L46-49)

Rancic, M., Purser, R. J., Pondeca, M., Lei, T., and Yokota, S.: Computational performance of the multigrid beta filter (MGBF) for covariance synthesis. Journal of Atmospheric and Oceanic Technology, in press, https://doi.org/10.1175/JTECH-D-24-0136.1, 2025. (L467-469)

**Response 2:**

Based on the chief editor's comment, we uploaded the initial and lateral boundary data and the unrestricted observation data on Zenodo, and revised the "Code and Data Availability" section as follows.

**Changes in manuscript:**

ICs, LBCs, and unrestricted observation data used in this study are obtained from https://doi.org/10.5281/zenodo.15744386, https://doi.org/10.5281/zenodo.15747449, and https://doi.org/10.5281/zenodo.15747476. The RRFS system used in this study is obtained from https://doi.org/10.5281/zenodo.15193112. (L344-346)

**Response 3:**

We added the following sentence in the "Acknowledgements" section at the funder's request.

**Changes in manuscript:**

This research of the National Centers for Environmental Prediction (NCEP) Environmental Modeling Center (EMC) is supported by NOAA's Science Collaboration Program and administered by UCAR's Cooperative Programs for the Advancement of Earth System Science (CPAESS) under award #NA21OAR4310383 and NA23OAR4310383B. (L360-363)

**RESPONSE TO REFEREE 1:**

We thank you for carefully reading our manuscript and giving useful comments. We revised the manuscript based on your comments. Our responses to your comments are described in the following, where your comments are italicized.

**Major comments 1**

In the original MGBF design (Purser et al., 2022), the filter is applied hierarchically across multiple resolutions ( $g_1$ ,  $g_2$ , ...,  $g_n$ ), with each level contributing to the final covariance operator. This multiscale construction is central to MGBF's ability to approximate broad localization functions and capture anisotropic or spatially inhomogeneous structures. The process involves adjoint and direct filtering at each grid level (see Eq. 18 and Purser et al., MWR 2022, p. 722), and the results are additively combined (Eqs. 16–17), ensuring smoothness, self-adjointness, and scalability.

In contrast, the present manuscript adopts a significant simplification: filtering is applied only at the coarsest filter grid, with no filtering at finer levels. This is a clear deviation from the original formulation, and although the authors mention it is for computational efficiency (Lines 99 and 306), the implications of this choice are not adequately discussed. Specifically, the manuscript should examine:

- How this approximation affects the effective shape of the localization function, especially for short localization length scales (e.g., 20 km);
- Whether it risks degraded performance (e.g., loss of sharpness or spurious correlations) in such cases;
- Whether the approximation is acceptable only in certain regimes, such as large-scale SDL with long localization radii, or whether it generalizes more broadly.

Clarifying these points would help readers understand the trade-offs and limitations of this modified implementation.

**Response:**

In MGBF-based localization, the interval of the filter grid should be smaller than the localization length. Therefore, the computational cost of the analysis with small localization length in MGBF is not necessarily smaller than that in RF. This limitation is both in single-scale localization and SDL. We explained this trade-off in the revised manuscript. The effect of the coarser filter grid with large localization length is negligible, which is shown as difference between MGBF00 (the filter grid is the same as the analysis grid) and MGBF04 (the filter grid is coarser than the analysis grid) in Fig. 4.

**Changes in manuscript:**

However, note that the computational cost of the analysis with small localization length in MGBF is not necessarily smaller than that in RF since the interval of the filter grid should be smaller than the localization length. (L329-331)

**Minor comments 1**

Line 55, The current title of Section 2.1, "Ensemble-variational (EnVar) data assimilation", does not reflect the fact that this subsection includes a detailed mathematical formulation of scale-dependent localization (SDL) as applied in the GSI-based 3DEnVar system. In particular, Eqs. (3) and (4) describe the decomposition of ensemble perturbations across multiple spatial scales and the corresponding block-structured localization matrix.

Since SDL is a significant methodological feature of the paper, both in terms of formulation and in experimental comparisons (e.g., RFSDL vs. MGBF04SDL), I recommend updating the subsection title to something more precise, such as "2.1 Ensemble-variational (EnVar) data assimilation with scale-dependent localization".

**Response:**

We added "with scale-dependent localization (SDL)" in the title of this subsection.

**Changes in manuscript:**

2.1 Ensemble-variational (EnVar) data assimilation with scale-dependent localization (SDL) (L56)

**Minor comments 2**

In Line 105, the manuscript states that interpolations are performed "from  $g_1$  to the analysis grid  $g_0$ ." Since  $g_1$  is referred to as the "finest filter grid," it may be misinterpreted as having equal or higher resolution than  $g_0$ . However, based on Table 2,  $g_1$  can in fact be coarser than the analysis grid (e.g., in MGBF03–04). I suggest the authors clarify the resolution relationship between  $g_0$  and  $g_1$  to avoid potential confusion.

**Response:**

As you pointed out, the finest filter grid  $g_1$  can be coarser than the analysis grid  $g_0$ . We explained it in the revised manuscript.

**Changes in manuscript:**

The finest filter grid  $g_1$  is the same as the analysis grid  $g_0$  or coarser. (L108)

**Minor comments 3**

In Line 137, the authors mention that the analysis grid resolution is twice as coarse as the FV3LAM model grid (i.e., 6 km vs. 3 km), but do not provide any justification or discussion of this design choice. Since this resolution difference could affect the representativeness or accuracy of of the ensemble background error representation, localization, and filter application (especially given the role of multigrid interpolation in MGBF), it would be helpful if the authors could clarify:

- The rationale for using a coarser analysis grid (e.g., computational efficiency, memory constraints, etc.),
- Whether this design introduces any limitations or trade-offs in terms of representativeness or localization sharpness,
- And whether the MGBF design is sensitive to the resolution mismatch between the filter grid and the model grid.

**Response:**

We adopted the coarser analysis grid to reduce the computation cost. As the limitation, it makes the resolution of analysis increments coarser and prevents to set the smaller localization length than the grid interval. We explained it in the revised manuscript. The interval of the analysis grid is sensitive to the optimal localization length, but it is equally sensitive in both RF and MGBF because the resolution of analysis increments does not depend on the filter grid.

**Changes in manuscript:**

The larger interval of the analysis grid reduces the computational cost but makes the resolution of analysis increments coarser and prevents to set the smaller localization length than the grid interval. (L143-145)

**Minor comments 4**

Table 2: The symbol "—" appears in several columns (e.g., "Number of the finest filter grids", "Weight of ( $g_1$ ,  $g_2$ ,  $g_3$ ,  $g_4$ )", filter specifications), but its exact meaning is not defined. It is unclear whether "—" indicates "not applicable," "not used," "same as previous case," or "no filtering applied." To improve clarity and reproducibility, I suggest the authors include a footnote or caption line in Table 2 to explicitly define what "—" represents in each context.

**Response:**

It indicates no filtering. We added the explanation in this table.

**Changes in manuscript:**

Weight  $(\omega_1, \omega_2, \omega_3, \omega_4)$  ("-" indicates no filtering) (Table 2)

**Minor comments 5**

Lines 248-250 and elsewhere: The sentence beginning with "Nevertheless, the difference from  $RF\cdots$ " is grammatically correct, but a bit hard to follow due to its length and repeated comparative structure. With multiple experiments and color-coded references mentioned together, the logical comparison becomes difficult to parse.

I suggest breaking it into two simpler sentences or rephrasing it for clarity. For example:

"MGBF04 $\sigma$  showed a smaller deviation from RF than MGBF04. Similarly, MGBF04 $\sigma$ SDL was closer to RFSDL than MGBF04SDL."

In fact, similar long and repetitive sentence constructions appear in several other places in the manuscript. I recommend that the authors go through the manuscript to revise such sentences for improved readability and flow.

**Response:**

Thank you for the suggestion to make the sentence clearer. We revised it as your suggestion.

**Changes in manuscript:**

Nevertheless, MGBF04σ (yellow line) showed a smaller deviation from RF (cyan line) than MGBF04 (pink line). Similarly, MGBF04σSDL (orange line) was closer to RFSDL (blue line) than MGBF04SDL (red line). (L255-256)

**Minor comments 6**

Figure 9: there seems to be a mismatch between the panel labels and their descriptions in the caption. Based on the plotted content, panels (a) and (b) appear to show RMSE and bias for temperature, while (c) and (d) show RMSE and bias for horizontal wind. However, the caption currently states that (a, c) are temperature and (b, d) are wind, which appears to be incorrect.

**Response:**

Thank you for pointing out the mismatch between the labels and the caption. We corrected it.

**Changes in manuscript:**

Vertical profiles of first guess departure (a,c) standard deviations (difference from RF) and (b,d) biases verified against assimilated in-situ observations [a,b: temperature (K); c,d: horizontal wind (m s-1)] in each cycling experiment (cyan: RF; pink: MGBF04; yellow: MGBF04σ; blue: RFSDL; red: MGBF04SDL; orange: MGBF04σSDL) from 15 UTC, 29 September to 00 UTC, 30 September 2022. (Figure 9)

**Minor comments 7**

Lines 305-306: The sentence "... and showed how to prevent the computational problem found in applying it" reads a bit awkwardly. The phrase "prevent the computational problem" is not the best fit here, since the issue already occurred during implementation.

**Response:**

We changed this phrase to "make the computation faster than the RF."

**Changes in manuscript:**

This study applied the MGBF for the ensemble covariance localization instead of the RF in the regional EnVar DA system, and showed how to make the computation faster than the RF. (L311-312)

**RESPONSE TO REFEREE 2:**

Dear Dr. Benjamin Ménétrier,

We thank you for carefully reading our manuscript and giving useful comments. We revised the manuscript based on your comments. Our responses to your comments are described in the following, where your comments are italicized.

**Major comments 1**

As mentioned by referee #1 in his/her major comment, this paper is a restrictive application of the full MGBF method. With only one filtering level left, equation 10 seems similar to the NICAS method I developed independently in 2020 (https://doi.org/10.5281/zenodo.4058620, and mentioned in equation 10 of https://doi.org/10.5194/gmd-15-7859-2022). However, the NICAS method can use adaptive unstructured subgrids, handle complex boundaries, and produce inhomogeneous and anisotropic localization functions.

**Response:**

As you pointed out, the idea to apply the filter with the coarse resolution is the same as the NICAS. While the NICAS applies a localization matrix on the unstructured coarse filter grid and interpolates it to the analysis grid directly, the MGBF-based localization applies a compact-support filter on the structured coarse filter grid and interpolates it from the coarsest filter grid  $g_T$  to the analysis grid  $g_0$  step by step. One advantage of the MGBF-based localization is high parallelization efficiency with the compact-support filter and the step-by-step interpolation. We explained it in the revised manuscript. It is technically possible to produce inhomogeneous and anisotropic localization functions in MGBF (Purser et al. 2022) although to show the impact is the future task.

**Changes in manuscript:**

The idea to apply the filter with the coarse resolution is the same as the Normalized Interpolated Convolution from an Adaptive Subgrid (NICAS) adopted in the Model for Prediction Across Scales-Atmosphere with the Joint Effort for Data assimilation Integration (JEDI-MPAS, Liu et al. 2022). The NICAS applies a localization matrix on the unstructured coarse filter grid and interpolates it to the analysis grid directly. On the other hand, the MGBF-based localization applies a compact-support filter on the structured coarse filter grid and interpolates it from the coarsest filter grid  $g_T$  to the analysis grid  $g_0$  step by step. One advantage of the MGBF-based localization is high parallelization efficiency with the compact-support filter and the step-by-step interpolation. (L323-329)

Liu, Z., Snyder, C., Guerrette, J. J., Jung, B.-J., Ban, J., Vahl, S., Wu, Y., Trémolet, Y., Auligné, T., Ménétrier, B., Shlyaeva, A., Herbener, S., Liu, E., Holdaway, D., and Johnson, B. T.: Data assimilation for the Model for Prediction Across Scales – Atmosphere with the Joint Effort for Data assimilation Integration (JEDI-MPAS 1.0.0): EnVar implementation and evaluation. Geoscientific Model Development, 15, 7859–7878, https://doi.org/10.5194/gmd-15-7859-2022, 2022. (L435-438)

**Major comments 2**

This kind of explicit convolution method on coarse subgrids is computationally efficient when the localization length-scale is large compared to the analysis grid cell size, since the subgrid can be coarse. However, I agree with referee #1 that it can become very expensive for smaller localization length-scales, because in this case a fine subgrid must be kept to maintain the localization function sharpness.

**Response:**

As you pointed out, the computational cost of the analysis with small localization length in MGBF is not necessarily smaller than that in RF since the interval of the filter grid should be smaller than the localization length. We explained this disadvantage in the revised manuscript.

**Changes in manuscript:**

However, note that the computational cost of the analysis with small localization length in MGBF is not necessarily smaller than that in RF since the interval of the filter grid should be smaller than the localization length. (L329-331)

**Major comments 3**

Another issue properly handled in the NICAS method and missing here is the localization normalization (i.e. diagonal coefficients of the localization matrix should all be equal to one). Figure 4 suggests that the MGBF method is perfectly normalized with all curves going to 1 at zero separation. However, I believe this is true only if the observation is located on a coarse grid node. Indeed, even if the continuous function  $B_p(x)$  is normalized (as mentioned after equation 11), the discrete low-resolution filters  $F_{BF}$  might not be, and even if they were, the final interpolation to then analysis grid would break this normalization. Only an outer diagonal scaling matrix taking all the operators (filters and interpolations) into account can ensure a proper normalization.

**Response:**

As you point out, the interpolation to the analysis grid slightly breaks the normalization, but the impact is negligible. The impact of the interpolation is shown as difference between MGBF00 (the filter grid is the same as the analysis grid) and MGBF04 (the filter grid is coarser than the analysis grid) in Fig. 4. The difference of the peak value is very small. We revised the description of the normalization to make it more accurate.

**Changes in manuscript:**

 $\mathbf{F}_{\mathrm{BF}(g_t)}^X$ ,  $\mathbf{F}_{\mathrm{BF}(g_t)}^Y$ , and  $\mathbf{F}_{\mathrm{BF}(g_t)}^Z$  denote isotropic line beta filters applied in each generation in x-, y-, and z-directions, respectively (self-adjoint  $N_{g_t} \times N_{g_t}$  matrices); for example, the filtering kernel of  $\mathbf{F}_{\mathrm{BF}(g_t)}^X$  is:

$$B_{p,t}(x) = c_{B,t}(1 - 4X^2)^p \quad \left(X \equiv \frac{|x|}{s\sqrt{2(2p+3)}} \le \frac{1}{2}\right)$$
(11)

where  $B_{p,t}(x) = 0$  in X > 1/2, the coefficient  $c_{B,t}$  is set to satisfy  $\int_{-\infty}^{\infty} [B_{p,t}(x)]^2 dx = \omega_t (\omega_t)$ : weight of  $g_t$ where  $\sum_{t=1}^{T} \omega_t = \omega_T = 1$ ), and s is the standard deviation of the self-convolution of  $B_{p,t}(x)$ , which is the

filtering kernel of
$$\mathbf{F}_{\mathrm{BF}(\mathbf{g}_t)}^X (\mathbf{F}_{\mathrm{BF}(\mathbf{g}_t)}^X)^T$$
 and can be shown to have the form:
$$B_{p,t} * B_{p,t}(x) = \omega_t (1-X)^{2p+1} \sum_{i=0}^p a_{i,p} X^{p-i} (1+X)^{2i} \quad (X \le 1).$$
 (12)

If we generalize the definition of binomial coefficients:

$$C(i,j) = \frac{i!}{(i-j)!j!},\tag{13}$$

then the coefficients can be expressed,

$$a_{i,p} = \sum_{j=0}^{\min(i,\lfloor(p-i)/2\rfloor)} \frac{C(p,j)C(p-i,2j)C(i,j)}{C(2p,2j)},$$
 where  $\lfloor \cdot \rfloor$  is the floor function. In the particular case,  $p=2$ , these coefficients are  $a_{0,2}=a_{1,2}=a_{2,2}=1$ .

The filtering kernel of  $\mathbf{F}_{\mathrm{BF}(\mathbf{g}_T)}^X(\mathbf{F}_{\mathrm{BF}(\mathbf{g}_T)}^X)^T$  obtained as the self-convolution of  $B_{2,T}(x)$  can be expanded as:

$$B_{2,T} * B_{2,T}(x) = (1 - X)^5 (1 + 5X + 9X^2 + 5X^3 + X^4) \quad \left( X \equiv \frac{|x|}{s\sqrt{14}} \le 1 \right),$$
(15)

where s is the standard deviation of  $B_{2,T} * B_{2,T}(x)$ . Unlike RF, s is smaller than the  $e^{-1/2}$ -folding scale  $\sigma$  in MGBF (here,  $s/\sigma \sim 0.92852$ ).

In MGBF, not only  $\mathbf{F}_{\mathrm{BF}(g_1)}^Z$  but also  $\mathbf{F}_{\mathrm{BF}(g_T)}^X$  and  $\mathbf{F}_{\mathrm{BF}(g_T)}^Y$  are parallelized for  $N_X N_Y$  because Eq. (11) is independently applied for each horizontal grid point only in the finite domain near the point. It indicates that communication between processors is limited to the exchange of halo grid points with spatially neighboring processors and all-to-all communication is not required in MGBF. (L109-129)

**Minor comments 1**

In section 2.1, equation (2) is already an approximation of the general 3DEnVar formulation. Indeed, the authors are using the same 3D localization matrix for all the auto- and cross-localization blocks between different analysis variables. This method is sometimes referred to as "Mark Buehner's trick" (used in https://doi.org/10.1175/2009MWR3157.1 clearly described and in section https://doi.org/10.1002/qj.2325). It assumes that all the analysis variables have roughly the same error correlation length-scale. Whether this assumption holds here or not, I think it should be mentioned.

**Response:**

As you pointed out, this formulation uses the same localization matrix in all analysis variables. We explained it in the revised manuscript.

**Changes in manuscript:**

In this formulation, the same localization length is applied to all analysis variables. (L66-67)

**Minor comments 2**

In equation (10) of section 2.3, the rightmost interpolation operator (D from g1 to gt) is actually not required if only one grid and one scale are used, as  $DD^T = I$ . If several grids are needed (e.g. g2 and g4 as in experiment MGBF03SDL), this interpolation operator is required to combine the scales with operator E, but the destination grid should be the finest grid used (here g2), not necessarily g1.

**Response:**

As you pointed out,  $\mathbf{D}_{g_T \leftarrow g_1}$  is not required in the single-scale localization. Actually, it was not calculated in the sensitivity experiments with single-scale localization in this study. We explained it in the revised manuscript. Even in SDL, it can be changed to  $\mathbf{D}_{g_T \leftarrow g_2}$  if the finest filter grid is  $g_2$ . However, this change to  $\mathbf{D}_{g_T \leftarrow g_2}$  hardly shortens the calculation time because of the load imbalance;  $g_2$  is calculated in the limited number of processors (see section 4d in Purser et al. 2022). Therefore, we adopted  $\mathbf{D}_{g_T \leftarrow g_1}$  in SDL.

**Changes in manuscript:**

Note that  $\mathbf{D}_{\mathbf{g}_T \leftarrow \mathbf{g}_1}$  is required only in SDL because  $\mathbf{D}_{\mathbf{g}_T \leftarrow \mathbf{g}_1} \mathbf{E} \mathbf{D}_{\mathbf{g}_T \leftarrow \mathbf{g}_1}^T = \mathbf{I}$  in single-scale localization. (L108-109)

**Minor comments 3**

Finally, I think that the experiments with slightly reduced length-scales (with a sigma suffix) are not really necessary. As shown in https://doi.org/10.1175/MWR-D-22-0255.1, the analysis quality is not very sensitive to the localization length-scale, as long as this length-scale is good enough. Given all the other uncertainties about the localization function shape and the fact that it should actually be anisotropic and inhomogeneous, the optimization of the localization length-scale does not seem really relevant here. Removing it (or better keeping it and removing the non-sigma case) would make the article a bit lighter and easier to read.

**Response:**

As you pointed out, the analysis quality in MGBF04 $\sigma$  was very similar to that in MGBF04. However, the difference of analysis quality was not negligible. Namely, MGBF04 $\sigma$  showed a smaller deviation from RF than MGBF04 (Figs. 6-9). Since this is one of the important conclusions of this study, we showed both MGBF04 and MGBF04 $\sigma$  and compared them. This result possibly implies that the analysis quality is occasionally sensitive to the localization length in the operational DA system unlike the idealized cases shown in Morzheld and Hodyss (2023).

**Changes in manuscript:**

(None)

**References:**

Purser, R. J., Rancic, M., and De Pondeca, M. S. F. V.: The multigrid beta function approach for modeling of background error covariance in the Real-Time Mesoscale Analysis (RTMA). Monthly Weather Review, 150(4), 715–732, https://doi.org/10.1175/MWR-D-20-0405.1, 2022.

Morzfeld, M., and Hodyss, D.: A theory for why even simple covariance localization is so useful in ensemble data assimilation. Monthly Weather Review, 151(3), 717–736, https://doi.org/10.1175/MWR-D-22-0255.1.

---

## Author Response (AR2)

**RESPONSE TO EDITOR:**

We corrected the following parts in the manuscript based on editor's comments.

**Comments 1**

line 144: "prevents to set the ...than the grid interval" -> "prevents setting a localization length smaller than the grid interval"

**Response 1:**

We corrected it as your comment.

**Changes in manuscript:**

The larger interval of the analysis grid reduces the computational cost but makes the resolution of analysis increments coarser and prevents to set the localization length smaller than the grid interval. (L143-145)

**Comments 2**

The NICAS method also uses compactly supported functions to improve the parallelization, which is not different from the MGBF. It is worthy to mention this or at least not to differentiate MGBF from NICAS on this point.

**Response 2:**

We corrected the explanation not to regard the compact-support filter as difference of MGBF from NICAS.

**Changes in manuscript:**

The idea to apply the compact-support filter with the coarse resolution is the same as the Normalized Interpolated Convolution from an Adaptive Subgrid (NICAS) adopted in the Model for Prediction Across Scales-Atmosphere with the Joint Effort for Data assimilation Integration (JEDI-MPAS, Liu et al. 2022). The NICAS applies a localization matrix on the unstructured coarse filter grid and interpolates it to the analysis grid directly. On the other hand, the MGBF-based localization applies a filter on the structured coarse filter grid and interpolates it from the coarsest filter grid  $g_T$  to the analysis grid  $g_0$  step by step. One advantage of the MGBF-based localization is high parallelization efficiency with the step-by-step interpolation. (L323-329)